# Diversity of Binge-Eating Disorder Symptoms Is Associated with Anxiety about Getting Fat Rather Than Body Image: A Clinical Study of Women in Poland

**DOI:** 10.3390/nu15214572

**Published:** 2023-10-27

**Authors:** Karolina Lewandowska, Waldemar Klinkosz, Wojciech Styk, Magdalena Kowalczyk

**Affiliations:** 1Institute of Psychology, The Cardinal Stefan Wyszyński University in Warsaw, 01-938 Warsaw, Poland; lewandowska.zola@gmail.com (K.L.); w.klinkosz@uksw.edu.pl (W.K.); 2Department of Psychology, Medical University in Lublin, 20-059 Lublin, Poland; 3Mental Health Clinic, Complex of Provincial Specialty Clinics in Katowice, 40-038 Katowice, Poland; m.lewand17@gmail.com

**Keywords:** binge-eating disorder, body image, body mass anxiety, anxiety about getting fat, overweight, obesity

## Abstract

Background: Anxiety about gaining weight is strongly related to body image. Evidence indicates that body dissatisfaction is a strong predictor of eating disorder development. Although not included in DSM-V diagnostic criteria, body image dissatisfaction, and concern are clearly relevant also for individuals with binge-eating disorder (BED). Weight gain anxiety is associated with psychopathological behaviors, but existing research in this area is primarily focused on bulimia nervosa and anorexia nervosa. The goal of this present study was to investigate body image and body mass anxiety in people with BED. Methods: Women diagnosed with BED (*n* = 105) aged 18 to 66 were surveyed using the questionnaire developed by the authors evaluating the presence of BED symptoms based on DSM-V criteria, and two other instruments: the Body Esteem Scale (BES), and the Body Mass Anxiety Scale (BMAS-20). Statistical analyses were conducted to examine the correlations of BED with body image and body mass anxiety (Pearson’s r), to test differences between groups with greater and lesser BED symptom diversity (Student’s *t*-test or the Mann–Whitney U test), and to assess differences among mild-, moderate- and severe-BED groups (ANOVA with a post-hoc test). Results: A medium positive relationship was found between anxiety about getting fat (AGF) and the diversity of BED, measured as the number of BED symptoms. A larger number of BED symptoms was shown to be associated with a higher level of AGF. However, no significant differences in AGF levels were observed among BED-severity groups, specified with the frequency of binge eating episodes. No correlations were found between BED and body image. There were also no significant differences in body image between groups with a larger and a smaller number of BED symptoms. The only significant difference in body image observed among BED-severity groups was the level of weight concern. People with mild BED displayed a higher level of weight concern than those with severe BED. Conclusions: Women who binge eat experience high levels of AGF. In the present study, AGF was primarily associated with the number of BED symptoms and not the rate of recurrence of binge-eating episodes. The frequency of BED episodes, however, was linked with weight concern. On the other hand, the hypothesized relationship between disturbed body image and BED was not confirmed. The findings indicate that anxiety about getting fat should be viewed as one of the psychological costs incurred by women experiencing BED symptoms, and it should be mentioned in the next DSM version.

## 1. Introduction

Binge-eating disorder (BED) involves recurrent episodes of uncontrolled eating without compensatory behaviors in order to control and reduce weight, such as self-induced vomiting, misuse of laxatives, or excessive exercise. A binge-eating episode occurs with at least three of the following symptoms: (a) rapid pace of eating; (b) eating until feeling uncomfortably full; (c) eating large portions of food despite not feeling physically hungry; (d) eating alone because of being embarrassed by how much one is eating; (e) feeling self-loathing, depression, or guilt after overeating. BED episodes lead to a high psychological burden, giving rise to guilt, disgust with one’s own body, self-devaluation, and general suffering connected with overeating [1]. The International Classification of Diseases ICD-11 defines the criteria for the diagnosis of BED in a similar way. In addition, it emphasizes that BED symptoms are not caused by any medical condition (e.g., Prader–Willi syndrome) or another mental disorder (e.g., bipolar disorder), and are not due to the effect of a substance or medication on the central nervous system [2]. According to the latest data, BED, as defined by the DSM-V, currently affects 1.5% to 1.9% of women and 0.3% of men worldwide [3]. The overall prevalence of BED in the world population is 0.9% [4].

There is a positive relationship between eating disorder symptoms and body image dissatisfaction and concern [5,6,7]. Moreover, according to Stice et al. [8], body dissatisfaction is the strongest predictor of eating disorder development. Although not included in DSM-V diagnostic criteria, body image dissatisfaction, and concern are clearly relevant also for individuals with BED [9]. Body image is a mental structure representing a person’s individual experiences related to the appearance of their own body [10]. It includes an individual’s perception of their body and emotional attitude toward it [11]. Several variables have been found to influence the relationship between body image dissatisfaction and eating disorders, including self-esteem, neuroticism, perfectionism, and body surveillance [6,12,13]. One way to assess the multifactorial construct of body image concerns is an overvaluation of shape and weight [14]. Various studies have emphasized the importance of overvaluation of shape and weight as a possible specifier for the diagnosis of BED [14,15]. In addition, Askew et al. [16] have demonstrated that preoccupation with body shape and weight is a predictor of binge eating. Furthermore, individuals with BED show significantly higher levels of body anxiety than obese individuals without a diagnosis of BED [17,18,19]. This allows one to, cautiously, formulate a hypothesis that body image dissatisfaction and concern are not exclusively due to overweight and obesity. This is confirmed by a study that showed that, regardless of their body weight, women with BED differed from healthy women in having a strong tendency to overestimate their body size [9].

One of the cognitive body image constructs is the fear of weight gain (weight gain anxiety) [17]. Styk et al. [20] have shown that weight gain anxiety is strongly negatively related to body image. They have demonstrated, using their own measure—the Body Mass Anxiety Scale—that both weight gain anxiety and weight loss anxiety could be associated with the fear of losing self-control. Interestingly, in some studies the fear of losing self-control has been found to be the most important predictor of eating disorders [21,22]. Previous studies showing a positive relationship between weight gain anxiety and psychopathological behaviors have focused mainly on bulimia nervosa and anorexia nervosa [23,24]. Styk et al. [20] have drawn attention to yet another aspect related to the anxiety experienced by patients with eating disorders, which is the anxiety about losing weight. This type of anxiety does not occur in specific clinical groups and it is much less prevalent; however, research conducted by those authors confirms the existence of this construct.

We expected that the severity of BED (measured as the frequency of binge eating episodes) and the diversity of BED symptoms (measured as the number of symptoms) would be negatively related to body image and positively related to body mass anxiety. We hypothesized that a greater severity and a greater diversity of BED symptoms would be associated with a more self-critical view of the body among the women surveyed, since recent data indicate that, regardless of body weight, women with BED have significantly reduced weight satisfaction compared to women without BED [9]. Moreover, a diagnosis of BED is associated with the Weight Concern and Shape Concern subscales on the Eating Disorder Examination 16.0 (EDE-16.0) [9]. We made a similar assumption for body mass anxiety, especially anxiety about getting fat. We expected that a greater severity and a greater diversity of BED symptoms would be associated with higher levels of AGF, as binge eaters tend to become overly preoccupied with their own weight [25].

## 2. Materials and Methods

### 2.1. Procedure

Participants were recruited through advertisements circulated in the following institutions providing treatment for overweight and obese patients and patients with eating disorders: Obesity Treatment Centre at Szpital [Hospital] pod Bukami in Bielsko-Biała, “Medispace” Medical Centre in Warsaw, Bariatric Clinic “Baria 3” in Wrocław, the all-Poland community of Anonimous Eataholics, and an Internet support group for people with binge-eating disorder “Wilcze Stado”. The surveyed women were patients of these institutions. The qualifying condition for people in the study group was, in addition to episodes of binge eating repeated at least once a week, the presence of at least three symptoms listed in the DSM-V criteria. A questionnaire developed by the authors based on DSM-V criteria (BED Symptom Questionnaire) was used to verify BED symptoms. To ensure a homogeneous sample, comorbid psychiatric diagnoses of bipolar disorder, and schizophrenia, were exclusion factors, as was pregnancy The study was approved by the Research Ethics Committee of the Institute of Psychology at the Cardinal Stefan Wyszyński University in Warsaw (registration number: 19/2022). The participants were informed of the purpose of the survey and survey procedures. They were also told that the survey was anonymous and that they could withdraw from participation at any time. They were asked to complete a set of questionnaires. The battery consisted of a personal data sheet, a BED Symptoms Questionnaire, the Body Esteem Scale (BES), and the Body Mass Anxiety Scale (BMAS). The respondents were divided into three groups based on the severity of BED symptoms. The severity of BED symptoms was measured as the frequency of binge-eating episodes: (1) mild—1 to 3 episodes a week; (2) moderate—4 to 7 episodes a week; and (3) severe—8 or more episodes a week. The diversity of BED symptoms was measured as the number of symptoms ticked by respondents in the BED Symptoms Questionnaire.

The body mass index (BMI) was calculated for each person surveyed on the basis of their body weight and height as declared in the personal data sheet. BMI (kg/m^2^) was calculated as weight/height^2^ and classified in accordance with the current WHO standards: underweight (BMI < 18.5 kg/m^2^); normal weight (BMI 18.5–24.9 kg/m^2^); overweight (BMI 25.0–29.9 kg/m^2^); obesity (BMI > 30.0 kg/m^2^) [3].

### 2.2. Measures

BED Symptoms Questionnaire. This questionnaire, developed by the authors, is strictly based on the diagnostic criteria of the American Psychiatric Association’s DSM-V classification of mental disorders. It is a self-report measure that covers eight BED symptoms and includes one control item regarding compensatory behaviors (item 9). The respondents’ task was to tick the symptoms that they had suffered from during the last month. The questionnaire allowed one to determine the diversity of BED based on the number of BED symptoms (from at least three to eight) and the general severity of BED based on the frequency of binge-eating episodes (mild: 1–3 times a week; moderate: 4–7 times a week; severe: 8 and more times a week). The symptoms included in the questionnaire are given in Table 1.

Body Esteem Scale (BES). BES is a self-report instrument that allows one to determine a respondent’s attitude toward their own body [26]. It comprises 35 items across three subscales, which are different for women (Sexual Attractiveness, Weight Concern, Physical Condition) and men (Physical Attractiveness, Body Strength, Physical Condition). Each item is rated on a 5-point Likert scale from 1 (I have strong negative feelings) to 5 (I have strong positive feelings), with 3 denoting a neutral attitude. The Polish adaptation of the test is characterized by high reliability, both regarding the subscales for women (Cronbach’s alpha from 0.80 to 0.89) and men (Cronbach’s alpha from 0.85 to 0.88).

Body Mass Anxiety Scale (BMAS-20) [20]. BMAS-20 is a self-report questionnaire that measures the level of weight-related anxiety. It contains 20 test items and consists of two scales: Anxiety About Getting Fat (AGF) and Anxiety About Losing Weight (ALW). Each item is rated on a 7-point scale from 1 (Does not concern me at all) to 7 (Fully concerns me). The tool has good psychometric properties—both the original Polish version (Cronbach’s alpha 0.916) and the English version (Cronbach’s alpha 0.91).

### 2.3. Participants

The study was conducted in a clinical group of 105 women aged 18 to 66 years (M = 33.76; SD = 9.59). The average BMI was 27.95; SD = 8.27. According to the criteria of the World Health Organization, six of the participants were underweight, 41 had a normal weight, 25 were overweight, and 33 were obese. A total of 67 women reported mild BED symptoms (1–3 episodes per week), 29 reported moderate symptoms (4–7 episodes per week), and nine reported severe symptoms (8 and more episodes per week). Eleven of the study participants were undergoing pharmacological or therapeutic treatment for BED. The participants were included in the sample if they met the DSM-V criteria for BED [1]. The presence of BED symptoms was assessed using the BED Symptoms Questionnaire, a DSM-V-based instrument developed by the present authors. In accordance with the diagnostic criteria, a person can be diagnosed with BED when they present at least three out of the eight symptoms listed in the DSM-V. Excluding factors such as bipolar disorder, schizophrenia and pregnancy were taken into account to ensure the sample was uniform.

### 2.4. Statistical Analysis

To test our hypotheses, we performed statistical analyses using JASP 0.17. We calculated basic descriptive statistics and ran the Shapiro–Wilk normality test. Correlation analysis was performed using Pearson’s *r*. The values of the variables were compared across the three BED-severity groups using Analysis of Variance (ANOVA), followed by Tukey’s post-hoc tests. In order to divide the participants into groups with a greater and lesser diversity of BED symptoms, a cluster analysis was performed. Next, the values of the test variables were compared between groups with a greater and lesser diversity of BED symptoms using Student’s *t*-test, or the Mann–Whitney U test when the assumptions of the parametric test were not met. The results were considered significant at α < 0.05.

## 3. Results

### 3.1. Descriptive Statistics and Correlation Analysis

First, we calculated the descriptive statistics of the results we had obtained. The normality of the distribution was assessed using the Shapiro–Wilk’s test. Table 2 shows the means and the standard deviations for the investigated variables.

As a next step, we conducted a correlation analysis. The results pointed to a medium positive correlation between AGF and the number of BED symptoms (*r* = 0.46; *p* < 0.001). Negative and slightly weaker correlations were observed between AGF and weight concern (*r* = −0.32; *p* < 0.001), as well as between sexual attractiveness (*r* = −0.29; *p* < 0.01) and the global body image score (*r* = −0.31; *p* < 0.01). ALW was not associated with any other variable. The results of the analyses are shown in Table 3.

### 3.2. Relationships of Severity of BED Symptoms with Anxiety about Getting Fat and Body Image

An ANOVA analysis of variance was performed to compare groups with mild, moderate, and severe BED symptoms. The median rate of symptoms in the first group was 6, in the second group—7, and in the third group—8. The results showed that the three groups differed significantly in BMI (*F*(2) = 10.43; *p* < 0.001), weight concern (*F*(2) = 3.42; *p* = 0.04) and number of BED symptoms (*F*(2) = 4.15; *p* = 0.02). The effect size for BMI was large, while the effect sizes for weight concern and number of BED symptoms were moderate [27]. To identify where exactly the differences occur, we conducted Tukey’s post-hoc test. The results showed that the group with severe BED (8 or more binge-eating episodes a week) differed from the groups with mild BED (1 to 3 binge-eating episodes a week) and moderate BED (4 to 7 binge-eating episodes a week) in terms of BMI. People with severe BED symptoms had the highest BMI (M = 38.78), *p* < 0.001. The mean BMI in this group was significantly higher than in the groups with moderate (M = 28.07), *p* < 0.001 and mild BED (M = 26.44), *p* < 0.001. In addition, the groups with mild and severe BED differed in weight concern and number of BED symptoms. People with mild BED displayed a significantly higher level of weight concern (M = 22.37), *p* < 0.05 than people with severe BED (M = 16.44), *p* < 0.05. Moreover, people with mild BED had significantly fewer BED symptoms (M = 5.66), *p* < 0.05 compared to those with severe BED (M = 7.33), *p* < 0.05. The remaining results did not point to any significant differences among the groups. The means, standard deviations, F values, and post-hoc comparisons for each group are shown in Table 4.

### 3.3. Relationship of Diversity of BED Symptoms with Anxiety about Getting Fat and Body Image

As a next step, we performed cluster analysis on the number of BED symptoms in the sample. A two-stage cluster analysis procedure was used: hierarchical cluster analysis and k-means analysis. Hierarchical cluster analysis was used to determine the optimal solution for the number of clusters. The clustering results of the hierarchical analysis suggested that two clusters would show significant differences and adequately represent the data. In the next step, a cluster analysis was performed using the k-means method with the number of clusters determined to be two. The results of the analyses of differences between the clusters are presented in Table 5.

Two groups emerged from the cluster analysis: group 1 with a median number of 7 symptoms (greater diversity of BED symptoms), and group 2 with a median number of 4 symptoms (lesser diversity of BED symptoms). The Mann–Whitney U test showed there was a statistically significant difference in the mean AGF between the study groups, *U* = 1848.50, *p* < 0.001. Women with a larger number of BED symptoms were, on average, characterized by a higher AGF (M = 56.45) than women with fewer symptoms (M = 39.57), *p* < 0.001. The effect size measured with the Glass rank biserial correlation coefficient was 0.64, which indicated the effect was large [28]. However, no significant differences were observed between the groups in AGF or body image (either the global BES score or BES subscale scores). The results of the analyses discussed above are presented in Table 5.

## 4. Discussion

The main aim of the present study was to investigate the level of anxiety about getting fat and the body image in women experiencing binge-eating episodes. We hypothesized that patients with BED felt a high level of body mass anxiety. Our results showed that the women with BED who participated in our survey were anxious that they would gain weight as an effect of binge eating. This observation is in keeping with the findings of a study by Lydecker et al. [25], who have demonstrated that binge eaters tend to be overly preoccupied with their body weight. There are studies that provide evidence that preoccupation with body weight is associated with eating disorders, [17] and may even be a predictor of them [16]. Askew et al. [16] (ibid.) have shown that self-esteem that is excessively affected by body weight and shape may predict binge eating.

In our study, we also observed a negative relationship between weight control and AGF, which indicated that women who believed they had more control over their body weight were less likely to get anxious about gaining weight. This result suggests that people with mild BED symptoms have a stronger self-control mechanism than those with severe BED. This is in line with Styk et al., [20] who have demonstrated that AGF is positively correlated with impulsiveness, which in the case of people with obesity may be a consequence of an interaction between high sensitivity to reward and poor self-control [29,30]. In addition, it can be assumed that, in some people, regular weight control may be used as a method of reducing anxiety about getting fat. However, further research is needed to confirm this assumption. Another finding of the present study was that AGF was negatively associated with sexual attractiveness and body image. The observation that women with a higher sense of being attractive and general satisfaction with their body were less likely to be anxious about gaining weight did not come as a surprise to us. This result finds confirmation in a study by Styk et al. [20], in which a strong negative relationship between body image and AGF was observed. In this present study, no association was found between BED and anxiety about losing weight. Although it is believed that ALW, similarly to AGF, is associated with the fear of losing self-control, our research did not confirm this relationship.

In our study, a significant difference in mean AGF was observed between the groups with a lower vs. higher diversity of BED symptoms. A larger number of BED symptoms (the greater diversity of BED) was shown to be associated with a higher level of AGF. However, no significant differences in AGF levels were observed among the BED-severity groups. Therefore, it can be assumed that the diversity of BED symptoms has more effect on the anxiety of getting fat than does the rate of recurrence of binge eating episodes. Also, it can be assumed, that the anxiety about getting fat has more impact on the diversity of BED symptoms than on the rate of recurrence of binge-eating episodes. In reference to the theory of addictions [31], it can be hypothesized that a larger number of BED symptoms may affect a larger number of areas of everyday functioning. The more symptoms, the greater the engagement in behaviors typical of BED. Preoccupation with binge eating can lead to the depletion of resources and, as a result, to the weakening of self-control. The sense of having lost control over eating is one of the diagnostic criteria for BED [2], while anxiety about getting fat is associated with the fear of losing self-control [20]. Our results lead to the conclusion that anxiety about getting fat should be viewed as another psychological cost incurred by people who experience eating binges.

No relationship between body image and the remaining variables investigated in the present study was observed. By the same token, the hypothesis that there is a relationship between body image dissatisfaction and the severity and diversity of BED symptoms was not confirmed by our results. The findings reported in the literature on body image in BED are inconsistent. Some studies have shown that obese people who have BED symptoms do not differ in body dissatisfaction from obese people without eating disorders [32,33]. Perhaps, a higher BMI has more impact on body dissatisfaction than does BED. On the other hand, other studies have found that binge eaters show greater body dissatisfaction compared to obese people who do not present with BED symptoms [17,19,34,35]. A few researchers have reported that a body image concern most commonly manifests as excessive concentration on body weight and body size [14,16,25]. Moreover, there is some evidence that people with BED display cognitive bias regarding the perception of their own bodies [14]. It is possible that this bias concerns some specific aspects of body image, such as preoccupation with body mass or overvaluation of body shape. The literature of the subject mentions four interrelated but conceptually distinct body image constructs—dissatisfaction with body weight and shape, overvaluation of body weight and shape, preoccupation with body weight and shape, and fear of gaining weight [17]. Studies have shown that these constructs are differently associated with different eating disorders, and so no one body image construct can capture clinical risk in eating disorders [16].

Unlike anxiety about getting fat, the diversity of BED symptoms most likely does not affect the evaluation of one’s own body. Regarding the severity of BED, the only significant difference in body image observed among the BED-severity groups was weight control. The medium effect indicates that people with mild BED (1–3 binge eating episodes a week) display a higher level of weight control than those with severe BED (8 or more episodes a week). This is confirmed by Ahrberg et al.’s study [36], in which people with BED were also observed to show an increased level of body checking and concern. Presumably, people who have greater control over their body weight display a greater level of self-control in general, and therefore are less likely to experience binge eating compared to people with weaker self-control. Moreover, it can be hypothesized that regular weighing minimizes the symptoms of BED and is a compulsive activity that reduces mental pain. Of course, this hypothesis requires further testing. One additional observation was that people with mild BED had significantly fewer BED symptoms compared to people with severe BED.

## 5. Limitations

Despite our efforts, certain limitations could not be avoided. The final number of people enrolled in the study was relatively small because we wanted to gather a homogeneous group of participants. Since the number of men who signed up for the study was too small, they had to be excluded from the sample altogether. As a result, the sample consisted solely of women, which increased the risk of gender bias. Further studies in a larger sample including male participants are needed to compare the relationships we investigated in terms of gender. Due to the small sample size, the BED-severity groups were not proportional. The cohort of people with severe BED was really exiguous and not homogeneous, since we had limited access to potential participants and since the number of patients with severe BED symptoms is generally lower than the number of patients with milder symptoms. An additional limitation was the fact that the items of the questionnaire developed by the authors for assessing the presence and severity of BED symptoms had not been standardized or normalized. However, we believe that the fidelity of the items to the DSM-V diagnostic criteria ensures the reliability of the data collected. Finally, in further studies, independent measurements of the participants’ height and weight should be made, to avoid the risk associated with the fact that people diagnosed with BED may have a disturbed body image and may tend to overestimate their body weight.

## 6. Conclusions

In the present study, we found that the diversity of BED symptoms, measured as the number of symptoms present, plays a greater role in the psychopathology of BED than the rate of recurrence of binge-eating episodes. These results regard the anxiety about getting fat, but the diversity of symptoms may also turn out to have an impact on other variables in the course of BED. To the best of our knowledge, no studies have been conducted so far that would take into account the diversity of BED symptoms as a potential diagnostic criterion. It can be assumed that the greater the number of BED symptoms diagnosed, the larger the number of areas of life they may affect, which means that a greater diversity of BED symptoms may worsen the overall quality of life. In addition, women suffering from BED were found to experience high levels of weight gain anxiety. Our findings indicate that anxiety about getting fat should be viewed as one of the psychological costs incurred by women experiencing BED symptoms, which is worth mentioning in the next version of the DSM.

## Figures and Tables

**Table 1 nutrients-15-04572-t001:** Questionnaire developed by the authors for evaluating BED symptoms.

Item	Polish Version	English Version
1	Powtarzające się sytuacje, gdy nie mogłem/mogłam opanować chęci jedzenia.	Repeated occurrences in which I could not control the urge to eat.
2	Jedzenie w dużo szybszym tempie niż zwykle.	Eating much more rapidly than normal.
3	Jedzenie aż do nieprzyjemnego uczucia pełności	Eating until feeling uncomfortably full.
4	Jedzenie dużych porcji jedzenia mimo nieodczuwania fizycznego głodu.	Eating large amounts of food despite not feeling physically hungry.
5	Jedzenie w samotności w poczuciu wstydu i zakłopotania.	Eating alone because of being ashamed and embarrassed.
6	Odczuwanie niechęci do siebie, obniżonego nastroju lub winy w trakcie/po napadzie objadania się.	Feeling disgusted with oneself, depressed, or guilty during/after overeating.
7	Poczucie braku kontroli nad jedzeniem podczas napadu objadania się.	Feeling out of control over eating during a binge-eating episode.
8	Cierpienie psychiczne związane z niepohamowanym objadaniem się.	Mental distress associated with binge eating.
9	Powtarzające się zachowania w poczuciu winy z przejedzenia, takie jak wywoływanie wymiotów/zażywanie środków przeczyszczających/intensywne ćwiczenia.	Recurrent behaviors caused by the guilt of overeating, such as purging/use of laxatives/intense exercise.
10	Jeżeli sądzisz, że występuje u Ciebie napadowe objadanie się, spróbuj określić jego częstotliwość w ciągu tygodnia: 1–3 razy w tygodniu 4–7 razy w tygodniu 8 lub więcej razy w tygodniu.	If you think you binge eat, try to say how often you binge eat during the week: A. 1–3 times a week B. 4–7 times a week C. 8 or more times a week.

**Table 2 nutrients-15-04572-t002:** Descriptive statistics and normality test.

Variables	M	SD	Sk.	Kurt.	S–W	*p*	Min	Max
Age	33.76	9.59	1.06	1.23	0.93	0.001	18.00	66.00
BMI	27.95	8.27	1.19	1.25	0.90	0.001	17.63	56.97
Number of BED symptoms (0–8)	6.01	1.92	−0.82	−0.16	0.88	0.001	1.00	8.00
Severity of BED symptoms	1.45	0.65	1.16	0.20	0.67	0.001	1.00	3.00
BES—global score	94.87	22.01	−0.12	−0.38	0.99	0.70	42.00	148.00
Sexual Attractiveness	39.90	7.76	−0.10	−0.31	0.99	0.68	20.00	59.00
Weight Concern	21.18	7.27	0.30	−0.47	0.97	0.02	10.00	43.00
Physical Condition	25.29	7.98	−0.07	−0.80	0.98	0.17	9.00	42.00
Anxiety About Getting Fat (AGF)	51.63	14.71	−0.79	−0.10	0.93	0.001	11.00	70.00
Anxiety About Losing Weight (ALW)	12.82	4.51	2.86	10.56	0.66	0.001	10.00	37.00

Analyses were performed for *N* = 105 observations; S–W—Shapiro–Wilk test; BES—Body Esteem Scale.

**Table 3 nutrients-15-04572-t003:** Pearson’s correlations between variables for the entire sample.

Variables	Variables							
	BMI	Number of BED Symptoms (0–8)	BES—Global Score	Sexual Attractiveness	Weight Concern	Physical Condition	Anxiety about Getting Fat (AGF)	Anxiety about Losing Weight (ALW)
BMI	-							
Number of BED symptoms (0–8)	0.03	-						
BES—global score	−0.36 ***	−0.09	-					
Sexual Attractiveness	−0.09	−0.05	0.82 ***	-				
Weight Concern	−0.40 ***	−0.13	0.88 ***	0.57 ***	-			
Physical Condition	−0.41 ***	−0.06	0.88 ***	0.58 ***	0.71 ***	-		
Anxiety About Getting Fat (AGF)	−0.01	0.46 ***	−0.31 **	−0.29 **	−0.32 ***	−0.18	-	
Anxiety About Losing Weight (ALW)	−0.14	−0.18	0.11	0.06	0.16	0.10	0.04	-

BES—Body Esteem Scale; ** *p* < 0.01, *** *p* < 0.001.

**Table 4 nutrients-15-04572-t004:** Comparison of BED symptom severity groups in terms of anxiety about getting fat (AGF) and body image (BES—global score and subscale scores).

Variables	Mild BED Symptoms (1) *n* = 67		Moderate BED Symptoms (2) *n* = 29		Severe BED Symptoms (3) *n* = 9		*F*(2)	*p*	η^2^	Post-Hoc
	M	SD	M	SD	M	SD				
BMI	26.44	6.36	28.07	9.45	38.78	9.64	10.43	0.001	0.17	1–2, 1–3
Anxiety About Getting Fat (AGF)	49.94	16.33	54.28	11.16	55.67	10.50	1.26	0.29	0.02	
Anxiety About Losing Weight (ALW)	12.67	4.04	13.41	5.93	12.00	2.18	0.43	0.65	0.01	
BES—global score	97.39	22.56	90.66	20.71	89.67	21.21	1.23	0.30	0.02	
Sexual Attractiveness	40.33	7.58	38.28	7.57	42.00	9.57	1.07	0.35	0.02	
Weight Concern	22.37	7.47	19.90	6.83	16.44	4.42	3.42	0.04	0.06	1–3
Physical Condition	25.88	8.20	24.48	7.64	23.44	7.70	0.57	0.57	0.01	
Number of BED symptoms (0–8)	5.66	2.08	6.41	1.40	7.33	1.32	4.15	0.02	0.08	1–3

Post-hoc analysis was performed using Tukey’s test; coefficients in bold indicate statistically significant relationships; BES—Body Esteem Scale.

**Table 5 nutrients-15-04572-t005:** Differences in body image and anxiety about getting fat between groups with a larger and a smaller number of BED symptoms.

Variables	Group 1 (Me = 7) *n* = 75		Group 2 (Me = 4) *n* = 30		Statistic	df	*p*	Effect Size
	M	SD	M	SD				
Age	32.84	9.42	36.07	9.77	869.50	103	0.07	−0.23
BMI	27.66	8.81	28.67	6.82	942.50	103	0.20	−0.16
Number of BED symptoms (0–8)	7	1.03	4	1.28	2214.00	103	0.001	0.97
BES—global score	94.25	22.02	96.40	22.28	−0.45	103	0.65	−0.10
Sexual Attractiveness	39.85	7.78	40.03	7.83	−0.11	103	0.92	−0.02
Weight Concern	20.81	7.28	22.10	7.27	−0.82	103	0.42	−0.18
Physical Condition	25.16	8.08	25.60	7.84	−0.25	103	0.80	−0.05
Anxiety About Getting Fat (AGF)	56.45	11.60	39.57	14.92	1848.50	103	0.001	0.64
Anxiety About Losing Weight (ALW)	12.49	3.26	13.63	6.69	1159.50	103	0.80	0.03

Note. The effect size statistics for Student’s *t*-test and the Mann–Whitney U test were Cohen’s d and the rank biserial correlation, respectively; coefficients in bold indicate statistically significant relationships; group 1—a larger number of BED symptoms, group 2—a smaller number of BED symptoms; BES—Body Esteem Scale.

## Data Availability

Not applicable.

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
