# Peer review of "Diversity of Binge-Eating Disorder Symptoms Is Associated with Anxiety about Getting Fat Rather Than Body Image: A Clinical Study of Women in Poland"

_nutrients, 2023, doi:10.3390/nu15214572_

Round 1
Reviewer 1 Report
Comments and Suggestions for Authors
Please see the attached file.

Minor editing of English language required.
Author Response
As suggested by the Reviewer, we corrected the title - the abbreviation BED has been expanded to its full name Binge Eating Disorder. We also added a geographical connotation of the country where the study was conducted.
Q1 In the Abstract (line 13), we used "binge eating disorder" in full with the abbreviation BED in brackets. The abbreviation is used later in the abstract to allow faster reading and understanding of the text. We did the same for the entire manuscript: the first time the term "binge eating disorder" was mentioned, we wrote it in full (line 43)
Q2 In the Conclusion (in the Abstract) we added the phrase about the significance of the findings to the next version of the DSM (line 36).
Q3 In Table 4 we added the age for each category (Mild, Moderate, and Severe). We have standardized DMS-V by using Roman numbers for the all manuscript.
Q4 Table 5 has been changed according to the Reviewer's suggestion.
Q5 In the Limitations, we changed the phrase (line 319) according to the Reviewer's suggestion
Q6 Our aim was for the theoretical part to be concise and to include the most important information. In addition, the topic of the paper covers a disorder that is poorly researched and has a limited number of sources. The references we have included are references to the works with the best methodological basis and the most frequently cited.
Reviewer 2 Report
Comments and Suggestions for Authors
The study described in the manuscript is interesting and relevant, but the paper will benefit from a major review, mostly regarding the description of methods and the results' presentation and discussion.
Please add a clear description of what was used as BED frequency, BED severity, number of symptoms, ... and the distinctions and overlaps between these in the Methods section. This issue makes several results very difficult to follow, as they are poorly defined and used differently throughout the text and tables.
For example, as BE episodes are used to assess BED severity (which is completely adequate!), please uniformize its use. This is particularly relevant in the abstract. Also, it should be clear wether the "diversity" of BE symptoms refer to their number or the results of the cluster analysis and subsequent analysis (please check also comment below).
Some issues regarding the analysis performed are unclear: (a) Was normality assessed using skewness or kurtosis or SW's test? (b) Pearson's correlation is adequate for variables with normal distribution; what was used when data presented non-normal distribution?
The major issue regarding the analysis refers to the cluster analysis. The sentence in lines 219-20 does not adequately describe the cluster analysis performed. This analysis should be clearly described in the methods, and some attention must be paid to the terms used to describe it.
Some other terms should be corrected: (a) Replace "self-developed questionnaire" with "questionnaire developed by the authors". (b) Do not use terms such as "moderately strong" correlation; instead, use a clear classification of effects.
Finally, the discussion may be improved regarding its structure. Despite I consider that this section focus all the major issues, these could be presented in a more systematic and structured way.
Comments on the Quality of English LanguageSome sentences need rephrasing and overall I suggest reviewing English language, but this issue does not condition the manuscript's readability.
Author Response
According to the Reviewer's suggestion, we added a clear description of the severity of BED (line 127), the diversity of BED (lines 130, 282, 283), and number of symptoms (line 130). Also, we added this information in the Abstract (lines 24 and 27).
We added the phrase about the normality of distribution assessed using Shapiro-Wilk’s test (line 238).
According to the Reviewer's suggestion, we replaced the term "self-developed questionnaire" with "questionnaire developed by the authors". The term "moderately strong" has been changed to "medium" (lines 23, 246), also "strong effect" to "large effect" (line 289), according to the classification of effects.
The structure of the discussion was systematized by dividing it into shorter paragraphs, which gave it a more clear structure.
Some of the variables analysed were not normally distributed, but the sample size was greater than 100. According to the guidelines for samples of N>100, parametric tests may be used [1].
As suggested by the reviewer, we have expanded the description of the cluster analysis (line 308-314).
- George, D.; Mallery, P. IBM SPSS Statistics 23 Step by Step A Simple Guide and Reference 14th Edition Answers to Selected Exercises; Routledge: New York, 2019;